# Three-Dimensional Guided Zygomatic Implant Placement after Maxillectomy

**DOI:** 10.3390/jpm12040588

**Published:** 2022-04-06

**Authors:** Nathalie Vosselman, Haye H. Glas, Bram J. Merema, Joep Kraeima, Harry Reintsema, Gerry M. Raghoebar, Max J. H. Witjes, Sebastiaan A. H. J. de Visscher

**Affiliations:** Department of Oral and Maxillofacial Surgery, University Medical Center Groningen, University of Groningen, Hanzeplein 1, P.O. Box 30.001, 9700 RB Groningen, The Netherlands; h.h.glas@umcg.nl (H.H.G.); b.j.merema@umcg.nl (B.J.M.); j.kraeima@umcg.nl (J.K.); h.reintsema@umcg.nl (H.R.); g.m.raghoebar@umcg.nl (G.M.R.); m.j.h.witjes@umcg.nl (M.J.H.W.); s.a.h.j.de.visscher@umcg.nl (S.A.H.J.d.V.)

**Keywords:** maxillectomy, guided surgery, zygomatic implants, digital, 3D, prosthetic rehabilitation, head and neck oncology, maxillary tumor, maxillary reconstruction, 3D VSP

## Abstract

Zygomatic implants are used in patients with maxillary defects to improve the retention and stability of obturator prostheses, thereby securing good oral function. Prosthetic-driven placement of zygomatic implants is even difficult for experienced surgeons, and with a free-hand approach, deviation from the preplanned implant positions is inevitable, thereby impeding immediate implant-retained obturation. A novel, digitalized workflow of surgical planning was used in 10 patients. Maxillectomy was performed with 3D-printed cutting, and drill guides were used for subsequent placement of zygomatic implants with immediate placement of implant-retained obturator prosthesis. The outcome parameters were the accuracy of implant positioning and the prosthetic fit of the obturator prosthesis in this one-stage procedure. Zygomatic implants (*n* = 28) were placed with good accuracy (mean deviation 1.73 ± 0.57 mm and 2.97 ± 1.38° 3D angle deviation), and in all cases, the obturator prosthesis fitted as pre-operatively planned. The 3D accuracy of the abutment positions was 1.58 ± 1.66 mm. The accuracy of the abutment position in the occlusal plane was 2.21 ± 1.33 mm, with a height accuracy of 1.32 ± 1.57 mm. This feasibility study shows that the application of these novel designed 3D-printed surgical guides results in predictable zygomatic implant placement and provides the possibility of immediate prosthetic rehabilitation in head and neck oncology patients after maxillectomy.

## 1. Introduction

Several reconstructive techniques are available for patients with complex defects of the mid-face and maxilla following tumor resection. The size and extent of the maxillary defect, patient factors, and comorbidities are decisive factors for the choice of surgical, prosthodontic, or combined rehabilitation after a maxillectomy. In cases when tumor resection has caused a relatively small maxillary defect, primary closure or surgical reconstruction with a local soft tissue flaps alone can lead to excellent functional and aesthetic results. For larger maxillary defects, reconstruction with a vascularized flap or prosthetic rehabilitation with an obturator prosthesis can be used, the latter remaining an important treatment in many institutions [1]. However, conventional obturator prostheses can have their drawbacks, mainly caused by lack of retention of the prostheses. Placement of endosseous implants in the native bone of the maxilla allow for improvement of retention of the obturator prosthesis and thereby increase the success of prosthetic rehabilitation. While there is often not enough bone volume for reliable implant placement, zygomatic implants can be used to improve the retention of the obturator prosthesis [1,2,3].

The literature reports good zygomatic implant survival rates (78.6 to 100%) after placing maxillary resections [4]. Primary implant placement at the time of ablative surgery along with early loading of implants has been shown to be an effective rehabilitation protocol [1,2,3]. Although the survival rates are promising, this more complex treatment modality is not a standard implant procedure among many clinicians. Due to drilling with long drills close to critical anatomical structures, compromised visibility, and for oncological cases, also the absence of anatomical landmarks, the oblique drill trajectories for placement of zygomatic implants are challenging [5]. Inaccurate placement could result in uncontrolled bleeding, damage to the orbit and its content, damage to the maxillary sinus, and traumatic fractures to the orbitozygomatic complex [6,7]. Moreover, inaccurate placement and angulation of the implant results in positional errors at the apex and of the prosthetic head. This possibly results in an undesired prosthetic outcome and may even make the use of the zygomatic implant unattainable.

Pre-operative 3D planning and guided placement and drilling according to a virtual surgical plan could solve these problems and result in lower risk of complications compared to the free-hand approach. With the use of virtual implant planning, an optimal inclination, position, and depth of the zygomatic implant can be chosen considering volume and anatomical variation of the malar bone [8]. Moreover, the ideal prosthetic platform positions can be planned, which eliminates the possible need for the intraoperative “guess work” involved with complex zygomatic implant rehabilitation [9].

While there is widespread experience in guided placement of endosseous dental implants and guided resection of tumors, a proper tool for guided placement of zygomatic implants in maxillectomy patients is not yet available. With the combination of the oblique bone surface, the long drill trajectories and the extent of the defects make designing guided templates a challenge. Any small angular or positional entrance error results in magnification of apical positional error at the tip of the drill [10]. The drill guide for zygomatic implant placement, introduced by Vrielink et al. [11] in 2003, which was solely based on available bone volume, unfortunately had an unfavorable accuracy. A technical note describing guided placement of zygomatic implants in atrophic maxillae lacks implant placement-accuracy analysis [12].

Recently, our group described a novel design of a fully digital 3D surgical planning for accurately executing the ablative surgery, placement of zygomatic implants, and immediate placement of an implant-retained obturator prosthesis in human cadavers [13].Therefore, the aim of this study was to assess whether this full 3D virtual workflow to guiding zygomatic implants placement and providing the patient with a printed surgical obturator prosthesis in head and neck cancer patients with a maxillary defect would be clinically feasible.

## 2. Materials and Methods

A total of 10 consecutive patients (7 female, 3 male, mean age of 66.3 years, range 45–73 years) who were treated for oral malignancies at the department of Oral and Maxillofacial Surgery at the University Medical Center Groningen were included. Patients either had a pre-existing defect of the maxilla (*n* = 3) or were scheduled for a maxillectomy (*n* = 7) with reconstruction an obturator prosthesis supported by zygomatic implants. All maxillary defects in this study are categorized as a class Brown IIb defect [14]. Patient, tumor, and defect characteristics are described in Table 1. For all patients, a complete 3D virtual surgical planning was made, in which zygomatic implants as well as an implant-retained obturator prosthesis were included.

### 2.1. Pre-Implant Procedure and 3D Planning

Prior to ablative oncological surgery, each patient underwent a diagnostic work-up consisting of both a CT and MRI of the head and neck region for ablative surgery and implant planning. In dentate patients, the natural dentition of dentulous patients was digitalized through 3D optical surface scanning and could be matched to the 3D patient models. In edentulous cases, additional cone-beam-computed tomography scan (CBCT) datasets of the patients wearing their conventional prostheses were obtained. The patient’s prosthesis was prepared before scanning: five radiopaque markers were added and spread over the prosthesis. Immediately after the scanning, a second scan of the prosthesis itself was performed. Through the radiopaque markers, the two CBCT-datasets of the patient and the prosthesis were merged to match the virtual prosthesis to the 3D models of the patient’s anatomy.

By using a multi-modality CT and MRI combined workflow for 3D resection margin planning [15], the tumor was delineated on the MRI data, after which this dataset was fused with the CT bone data in order to construct a 3D bone and tumor model. This model enabled reliable virtual resection planning with oncologic margins [16]. The virtual patient dentition or prosthesis was matched to the virtual planning to allow for digital obturator prosthesis designing, matching the defect, and backwards planning of the zygomatic implants from the position of the obturator prosthesis. The zygomatic implant heads were placed in the most ideal prosthodontic positions. The apical part of the zygomatic implant was planned in the lateral cortical bone of the zygomatic complex with care for maximal bony contact of the implant. The needed length of zygomatic implant was determined. In dentate cases, two zygomatic implants were digitally planned at the maxillary defect site. Four zygomatic implants were planned in edentulous cases.

### 2.2. Guide Design

Translation of the 3D VSP towards the surgical procedure was realized by means of 3D-printed surgical guides (Figure 1).

Subsequently, patient-specific implant drill guides were designed based on the preferred apical and abutment positions of the zygomatic implants captured in the final virtual set-ups (3-Matic Medical, Materialise, Leuven, Belgium). In edentulous cases, the drill guides were developed to fit the alveolar ridge, nasal aperture, and zygomatic arch for stable positioning (Figure 2A,B).

The maxillary bone-supported part included an extension to the nasal aperture to verify correct positioning of the guide [17] and was connected with crosslink arms to the zygomatic bone-supported part. Centered channels in the drill-guides enable insertion of stainless steel milled drill sleeves, which should minimize deviation of the drill trajectories and prevent polyamide particle formation (Figure 2C,D). The length of the channels functions as an integral depth stop for the zygomatic implants (Figure 2E,F).

In addition, the guide was supplied with holes for temporary fixation with mini screws. If natural dentition was remaining after resection, the teeth were used for support of the guides (Figure 3). The guides were printed from polyamide, produced according to the ISO 13485 standards for medical devices, by Oceanz (Ede, The Netherlands).

### 2.3. Surgical Procedure

First, the tumor was removed by resecting the maxilla (SV) according to the preplanned, individually designed cutting guides (Figure 3A,B). In the two cases in which the maxillectomy already had been performed, a mucoperiosteal flap was raised. Second, the zygomatic implant drill guide was fitted onto the bone. All zygomatic implants were placed by the same surgeons (S.V. and G.R.). During exposure of the maxillary and zygomatic bone, care was taken in order to remove all connective tissue from the guide supporting bone region so that the drill guide could be seated with a tight fit. The guide was fixated with osteosynthesis screws (KLS Martin, Tuttlingen, Germany) (Figure 3C). Third, the first metal sleeves matching the 2.7 mm zygomatic drill with apical lance were inserted in the guide to create the entry point in the malar bone. Subsequently, the preplanned drill trajectories were performed (Figure 3D). The metal sleeve was removed, which transformed the guide into a placing guide for the correct installation angle for the zygomatic implants (Zygex, Southern implants, Gauteng, South Africa). Next, the implants were inserted into the zygomatic bone until the fixture mounts contacted the reference stop on the guide. (Figure 3E). Due to longitudinal slots in the guide, the guide can be removed easily following implant placement by loosening the osteosynthesis screws and unclipping the guide from the implants (Figure 3F). Before removing the guide, the maxillofacial prosthodontist determines the final screw direction of the fixture mount, which corresponds exactly with the abutment position (Figure 3G). The obturator prosthesis with preplanned slots can be used as a reference to ensure a parallel positioning of the prosthetic platforms. In the edentulous cases (*n* = 5), a second guide was placed on the contralateral side, and the guided implant procedure was repeated. The surgical procedure was finalized by fixating the obturator prosthesis. Non-engaging prosthetic cylinders (Southern implants, Irene) were fixed to the obturator prosthesis with ultraviolet light-curing resin. The obturator prosthesis was checked for balance support and was firmly screw-fixed on the zygomatic implant abutments (Figure 3H). The screw-retained retention allows post-operative removal of the surgical obturator prosthesis and enables replacement as often as necessary

### 2.4. Analysis of Accuracy

All patients underwent a routine postoperative cone-beam-computed tomography scan (CBCT) within 16 days after surgery, which was used to evaluate the accuracy of the implant placement. The computer-aided design (CAD) files in STL format of the titanium zygomatic implants were superimposed onto the postoperative CT data, and a comparison was made with the planned positions by calculating reproducible reference planes in which the accuracy was measured. The implant coordinate system (ICoS) includes three reproducible reference planes in which the accuracy was measured: the center of the zygomatic implant head, bone entry point of the implant, and bone exit point of the implant (Figure 4A). Furthermore, the 3D angular deviation between 3D-planned position and postoperative implant position was calculated (Figure 4B).

Deviation of abutment position in two dimensions were calculated by defining a plane parallel to the prosthetic occlusional plane as reference: the occlusion plane coordinate system (Figure 4C). If the implant-retained obturator prosthesis on the zygomatic implant abutments was within 3 mm of the prosthetic cylinders in the slots, and a passive fit could be achieved, placement was deemed a success.

## 3. Results

### 3.1. Implant Placement Accuracy

The surgical guides fitted well in 9 cases (28 zygomatic implants). In one case, the fit of the surgical guide was not optimal because a larger resection of the tumor than planned was performed. These two implants were placed non-guided and therefore eliminated from the accuracy analysis. The implant lengths varied between 35 mm and 55 mm and were placed with a mean entry point deviation of 1.73 ± 0.57 mm and a 3D angle deviation of 2.97 ± 1.38° (range 0.6–6.1°). The 3D accuracy of the abutment positions was 1.58 ± 1.66 mm. The accuracy of the abutment position in the occlusal plane was 2.21 ± 1.33 mm, with a height accuracy of 1.32 ± 1.57 mm. An overview of the accuracy results can be seen in Table 2 and Table 3. The accuracy was well within tolerance limits.

### 3.2. Fit of the Implant Retained Obturator Prosthesis

In nine cases, the obturator prostheses could be fixated with non-engaging prosthetic cylinders (Zygex Southern implants, Gauteng, South Africa) to the zygomatic implants as planned. The prosthetic outcome in the horizontal and vertical dimension was within the 3 mm leeway space. This margin was available in prepared slots of the obturator prostheses needed for fixation. In the case where the zygomatic implants were not guided placed, extensive prosthetic adjustments at the preplanned slots were needed to allow for a proper fit of the obturator prosthesis.

Finally, all pre-operatively designed obturator prostheses had an adequate and were well-balanced on the zygomatic implants and remaining maxillary structures.

## 4. Discussion

This feasibility study shows that the application of 3D-printed surgical guides results are feasible in predictable zygomatic implant placement and immediate prosthetic rehabilitation in head and neck oncology patients after maxillectomy. Furthermore, application of this reliable method is believed to minimalize the risk of surgical and prosthetic complications.

The literature reports loading of zygomatic implants within a few hours after implant placement [18,19], but to the best of our knowledge, such a CAD workflow involving immediate implant-retained prosthetic rehabilitation in a combined surgical procedure with guided tumor resection and placement of zygomatic implants is not described. Thereby, comparative accuracy data are not available yet. Perioperative prosthetic delivery obviates invasive impression taking in surgical field or shortly after surgery, which is a direct benefit for the patient.

In the literature, an unfavorable zygomatic implant position of the apex or prosthetic head is described as a surgical complication [2,7]. This could indicate that even when executed by experienced surgeons, there is a frequent occurrence of suboptimal zygomatic implant positioning using a free-hand placement. The concept of guided zygomatic implant placement was first tested by our group in a series of human cadavers [12]. The data of this pre-clinical cadaver study and the data presented here are comparable in accuracy. As a consequence, immediate implant support was available for the obturator prosthesis.

This phase I trial shows high clinical potential for this approach of 3D-planned placement of zygoma implants. We think that a larger group of patients is required to confirm our first data on the predictability of placement and subsequent immediate loading of the obturator prosthesis. The lessons learned from this trial are that 3D planning can be accurately used when surgeons and prosthodontists together plan the surgery and prosthetic rehabilitation. 3D visualization of the tumor and planned resection promotes clinical debate and facilitates choices. The execution of the resection is less of a determining factor. Added resections are very well possible since the support for the 3D zygoma guides are chosen outside the expected oncological surgical field. Two factors are critical for accurate placement of the zygomatic implants. The first is the accurate placement of the 3D guide. Surgeons should be aware how the guides should be placed and 3D information should be available in the OR. Time must be taken to place these correctly, as is the case with all 3D-planned surgical guides for another purposes. Second, during placement of the implants, the surgeon should have the possibility of visual inspection of the entry point in the zygoma. Despite accurate 3D planning and well-thought out guide design, the surgeon needs visual feedback on the entry point. Once the entry point is placed accurately, the rigid guide supports the right direction of the implant drill.

Besides guided placement of implants, currently, implant placement using real-time navigation is gaining popularity. Research results are promising, and this most likely accurate and less-invasive surgical technique could be a next step in zygomatic implant placement according to a VSP in the future. To date, the main drawback of current visualization techniques is the difficulty of steadily maintaining the drill handpiece and transferring the surgical view from the navigation display to the operative site, which is amplified in the long drills used for zygomatic implants [20]. Secondly, it currently involves above-average operating time [9].

It is reasonable to assume that knowledge of the planned resection automatically provides 3D visualization of the necessary obturator outline to restore oral function. In this study, a treatment protocol was used for immediate prosthetic rehabilitation with immediate loading of the zygomatic implants. Restoring oral function immediately after ablative surgery obviates the need for fitting, placing, and adapting the prostheses. After maxillectomy, the frequent necessity of adjuvant radiotherapy limits the possibility of achieving sufficient retention for a conventional obturator prosthesis. An implant-retained obturator prosthesis allows for repeated removal to check the oncological defect visually or in the event of complications. The addition of subsequently placing a fixed, removable obturator prosthesis during surgery is a major step to shortening the time of prosthetic delivery and implant utilization. It can be anticipated that the number of prosthetic interventions post-operatively will be less compared to conventional prosthetic planning, in which retention is more difficult to obtain. We anticipate that oral function in such patients can recover earlier and better before the often necessary radiotherapy starts, and the hospital visits for prosthetic aftercare will be minimized in the early post-operative phase. In case of adjuvant radiotherapy, it is important to provide zygomatic implant-specific information to the radiotherapy team. This enables adjustments of the radiotherapy treatment plan and the dosimetric accuracy in radiotherapy [21,22].

## 5. Conclusions

A fully digitalized workflow for guided resection, zygomatic implant placement, and immediate prosthetic rehabilitation is feasible when planning a zygomatic implant-retained prosthesis. The method presented here is novel and advantageous for head and neck cancer patients because of an immediate implant-based prosthetic rehabilitation after ablative surgery, which otherwise could not have been achieved without delay.

## Figures and Tables

**Figure 1 jpm-12-00588-f001:**
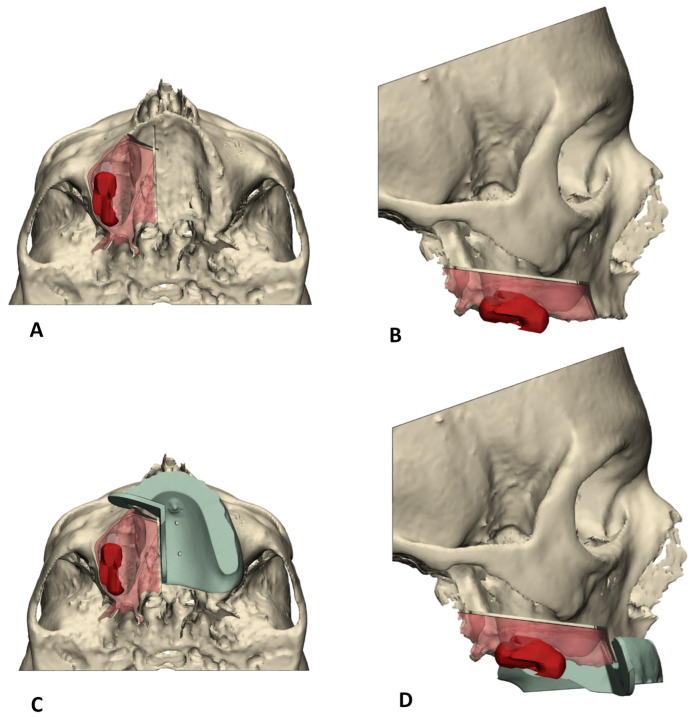
Overview of 3D VSP workflow for virtual resection planning. The working method starts with right-sided maxillectomy (**A**), caudal view (**B**), and the matching lateral view guided by the surgical cutting guides (**C**,**D**), with which the aim is to remove the red transparent part representing tumor removal with margin.

**Figure 2 jpm-12-00588-f002:**
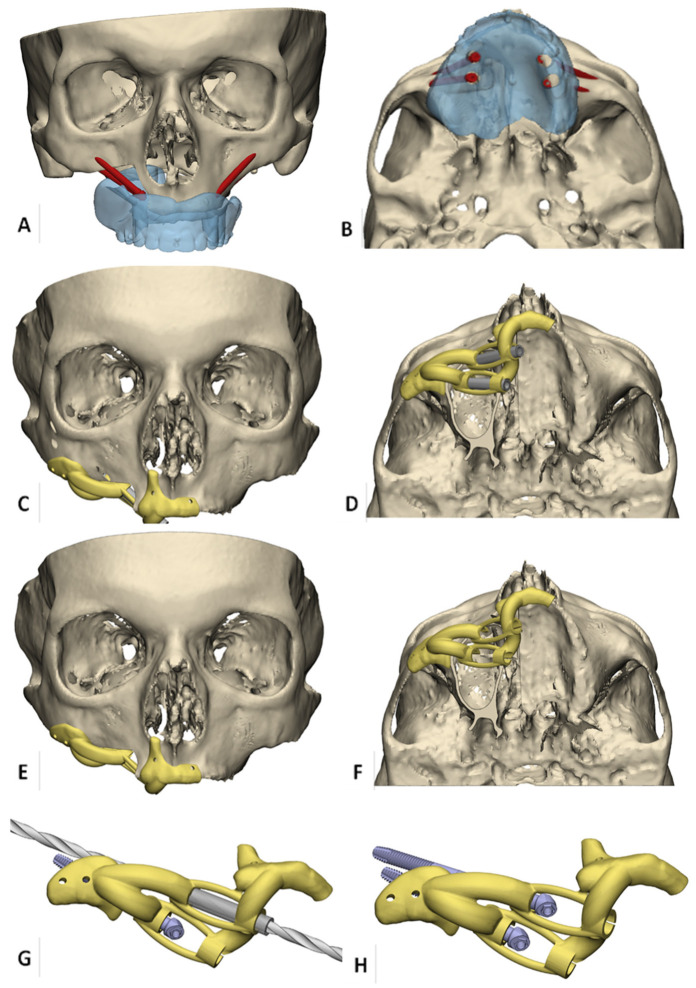
Overview of 3D VSP workflow for virtual zygomatic implant planning. (**A**,**B**) Virtual obturator prosthesis-driven zygomatic implant planning in an edentulous patient with right-sided maxillary resection planning. (**C**,**D**) Bone-supported zygomatic implant drill guide. Support is gained at alveolar ridge, nasal aperture, and zygomatic arch for stable positioning, and centered channels in the drill-guide enable insertion of stainless steel milled drill sleeves. (**E**,**F**) the length of the channels forms an integral depth stop for the zygomatic implants. (**G**) detailed view of drill guidance. (**H**) detailed view of zygomatic implants placed through the guide to enhance correct prosthetic head positions.

**Figure 3 jpm-12-00588-f003:**
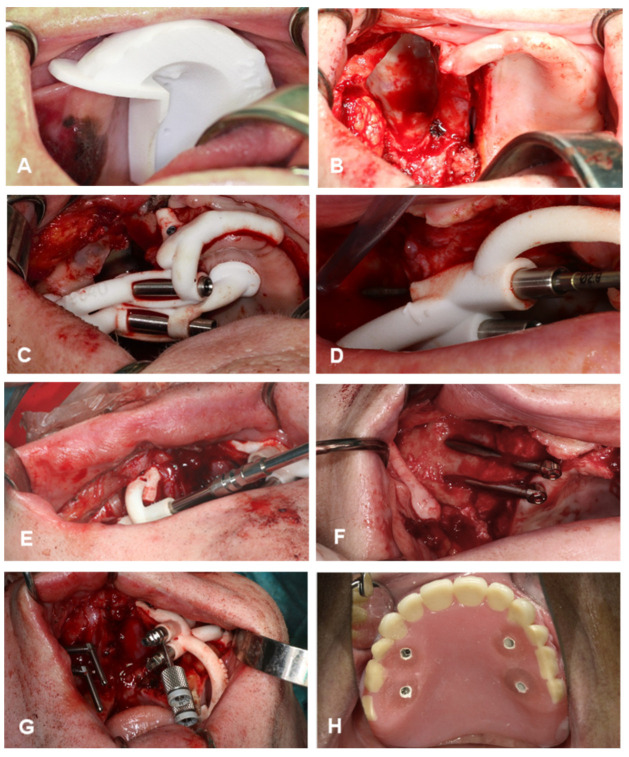
Surgical procedure. (**A**,**B**), maxillectomy according to the preplanned, individually designed cutting guides. (**C**) Drill guide seated with a tight fit and fixated with osteosynthesis screws. (**D**) Zygomatic drill inserted in the guide to perform the preplanned drill. (**E**) Insertion of zygomatic implant into the bone until the fixture mount contacts the reference stop on the guide. (**F**) View of zygomatic implant positions after removing the guide. (**G**) Final screw direction of the fixture mounts, which correspond exactly with the abutment positions. (**H**) Implant-retained obturator prosthesis immediately fixated with non-engaging prosthetic cylinders mounted into the prepared slots with a light-curing denture resin.

**Figure 4 jpm-12-00588-f004:**
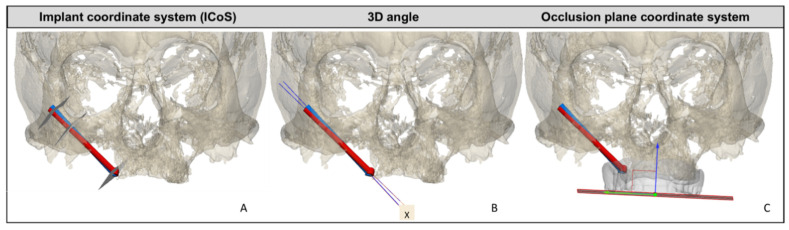
Overview of the different types of measurements and reference planes or coordinate systems for assessing the accuracy of zygomatic implant placement derived from post-op CBCT. In red, the planned zygomatic implant position; in blue, the postoperative zygomatic implant position. (**A**) The implant coordinate system (ICoS), including the three reproducible reference planes in which the accuracy is measured: the center of the zygomatic implant head, bone entry point of the implant, and bone exit point of the implant. (**B**) 3D angular deviation between 3D-planned position and postoperative implant position. X represents the 3D angle deviation. (**C**) Occlusion plane coordinate system. A plane parallel to the prosthetic occlusional plane is defined; perpendicular to this plane is the blue arrow. This arrow indicates the direction in which the abutment height accuracy is calculated. Deviation of the abutment is measured in the occlusional plane (green arrow).

**Table 1 jpm-12-00588-t001:** Patient, tumor, and defect characteristics.

Patient	Age (Years)	Sex	Indication	Laterality	Implants	IMPL Length (mm)	Radiotherapy
1	49	F	cT4N0 Adenoid cystic carcinoma maxilla	R	2	42.5; 55	Post-op
2	73	F	cT1N0 Squamous cell carcinoma maxilla	R & L	4	52.5; 45; 52.5; 47.5	Pre-op Post-op
3	64	F	cT4aN1M0 Squamous cell carcinoma maxilla	R	4	55; 50	-
4	74	M	pT4aN0M0 Melanoma cavum nasi	R	2	55; 55	Post-op
5	71	F	cT3N0M0 Oral lentiginous melanoma maxilla	R	4	35; 45; 42.5; 50	Post-op
6	67	M	T4N0 Squamous cell carcinoma maxilla	L	2	47.5; 55	Pre-op
7	60	F	cT4N0 Squamous cell carcinoma maxilla	R	2	47.5; 55	-
8	45	M	Langerhans Histiocytosis	R & L	4	55; 52.5; 55; 52.5	Pre-op
9	66	F	Osteosarcoma maxilla	R	2	45; 50	-
10	71	F	pT4aN0 Squamous cell carcinoma maxilla	R&L	4	55; 50; 55; 47.5	-

**Table 2 jpm-12-00588-t002:** Accuracy data. Result of the post-operative analysis of the implant coordinate system (ICoS) measurements. * SD, standard deviation.

ICoS Measurements *n* = 10	Mean (+/− * SD)	Min	Max
Abutment (mm)	1.60 (+/−0.64)	0.53	3.42
Entry point (mm)	1.81(+/−0.64)	0.43	3.24
Exit point (mm)	2.87 (+/−1.18)	1.11	4.72

**Table 3 jpm-12-00588-t003:** Accuracy data. Result of the post-operative analysis. Descriptive statistics of the occlusion coordinate system (OCoS) measurements. * SD, standard deviation.

OCoS Deviations *n* = 10	Mean (+/− * SD)	Min	Max
Abutment in occlusal plane (mm)	2.45 (+/−1.35)	0.87	6.04
Abutment height from occlusal plane (mm)	1.58(+/−1.66)	0.01	6.58
Axial angle (°)	2.31 (+/−1.52)	0.19	4.34
Coronal angle (°)	2.43 (+/−1.73)	0.25	7.97
Sagittal angle (°)	2.85 (+/−1.88)	0.27	7.04
3D angle (°)	3.20 (+/−1.49)	0.34	6.13

## Data Availability

The data presented in this study are available in Table 2. The raw data presented in this study are available on request from the corresponding author. Requests for materials should be addressed to N.V.

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
