# Peer review of "Three-Dimensional Guided Zygomatic Implant Placement after Maxillectomy"

_jpm, 2022, doi:10.3390/jpm12040588_

Round 1

Reviewer 1 Report

The manuscript is nicely written. I have only one suggestion which the authors should address in the discussion. The placement of zygomatic implants intra- operatively may complicate planning and delivery of post-operative radiotherapy. 

I find the manuscript suitable for publication.

Author Response

Response to Reviewer 1 Comments

Point 1: The manuscript is nicely written. I have only one suggestion which the authors should address in the discussion. The placement of zygomatic implants intra- operatively may complicate planning and delivery of post-operative radiotherapy. I find the manuscript suitable for publication.

Authors: Thank you for your comments and for the suggested improvements. Your suggestion is addressed in the manuscript.(line number 289-292).

Reviewer 2 Report

hello

thank you for a very interesting paper. It's good, but since my struggle with lots of patients after maxillectomy I would want to know, and highlight some topics:

  • what about brown - shaw maxillary defect cetegorisation? does it matter in this case studies?
  • is it possible to maintain oncological margins R0 in this 3d planning and how? perhaps some intra-operative histopathology? others?
  • how is it possible to evaluate oncological follow-up and estimate good healing under the fixed zygomatic implant prosthesis? is the patient able to remove the prosthesis by himself (for hygiene, for self-examination, others.,..), or every time an experienced surgeon is removing the appliance to evaluate the post-excisional cavity?
  • how exactly this 3d planning for surgery and reconstruction might improve very important factors for each patient, such as?: achieving proper soft/hard palate continuity to improve speech and swallowing; possibility to clean and maintain good hygiene of the excisional cavity (ex: mucus fluid accumulation from sinuses); support for soft tissues of cheeck/lips?'; does the material used in preseted cases/patients is in all cases hypoalergic and doesnt cause any irritations or minor./major injuries towards surrounding soft tissues, especially that RTH-therapy pre/post surgery always cause some serious mucosities and related soft tissues responses which require food hygiene and follow-ups; is there any connection with good prosthesis adherance on zygomatic impalnts to cover the nasal floor defect and improve speech and nasal breathing?
  • is there any realtion between rth-thearpy post/pre operation and the condition of zygomatic implant placement
  • in my surgical opinion, those serious factors are always necessary for patients and clinicians, especially each maxillectomy is a devastating procedure, nevertheless immediate implant placement after3d planning is always good for the patient for esthetical, functional and even psychological reasons
  • Regardless, nice topic, intresting study, recquiring a little bit....

Author Response

Authors:

Thank you for your comments. In response to your comments on our feasibility study we want to respond and inform you about the additions we made in a point by point fashion;

1 what about brown - shaw maxillary defect categorisation?

Thank you for your comment. The focus of this study is to describe reliable placement of zygomatic implants by 3D VSP and guided surgery. However, a reference to the Brown classification of the resulting maxillary defects in this study is missing and added in the manuscript. (line number 85,86).  

2 is it possible to maintain oncological margins R0 in this 3d planning and how? perhaps some intra-operative histopathology? others?

In our workflow we make a complete resection planning including 3D tumor visualization to determine the osteotomy planes with an oncologic margin (as the tumor is visualized on this model extracted from MRI/CT) . We previously described this validated workflow in a paper. We added reference [16] to support our statements.

3 how is it possible to evaluate oncological follow-up and estimate good healing under the fixed zygomatic implant prosthesis? is the patient able to remove the prosthesis by himself (for hygiene, for self-examination, others.,..), or every time an experienced surgeon is removing the appliance to evaluate the post-excisional cavity?

Thank you for your comment. The focus of this study is to describe reliable placement of zygomatic implants by 3D VSP and guided surgery.

 However, to further elucidate the procedure, we added a sentence to clarify the possibility to remove the surgical obturator prosthesis and screw it back in place again as often as necessary. Changes are made and  the manuscript is corrected. (line number 177-181).

4 how exactly this 3d planning for surgery and reconstruction might improve very important factors for each patient….

Thank you for your comment. The focus of this study is to describe how we can achieve accurate and reliable placement of zygomatic implants by 3D VSP and guided surgery. Because this was one of the main drawbacks of the use of zygomatic implants as they were often misaligned and complicated prosthetic rehabilitation. We tried to show in this paper that we can overcome this problems by our presented technique.

However, the clinical challenges you address in your  question are certainly very relevant.

We will present this in a future follow up paper (clinical outcomes one year after zygomatic implant placement and prosthetic rehabilitation). At the moment we have the impression that the technique helps to overcome prosthetic difficulties in the early postoperative phase after maxillectomy and during the often necessary radiotherapy. It appears that prosthetic complications such as  lack of retention, leakage of fluids are less compared to conventional prosthetic obturation.

5 is there any realtion between rth-thearpy post/pre operation and the condition of zygomatic implant placement.

I is favorable to place the zygomatic implants in non-radiated bone. This technique of 3D implant planning and placement potentially allows reliable and accurate placement in bone outside the radiation fields. In our future paper we will address to this important topic.